# In Silico Examination of Single Nucleotide Missense Mutations in *NHLH2*, a Gene Linked to Infertility and Obesity

**DOI:** 10.3390/ijms24043193

**Published:** 2023-02-06

**Authors:** Allison T. Madsen, Deborah J. Good

**Affiliations:** Department of Human Nutrition, Foods and Exercise, Virginia Tech, Blacksburg, VA 24060, USA

**Keywords:** nescient helix-loop-helix 2, NSCL-2, basic helix-loop-helix, transcription factor, tertiary structural analysis, Prader–Willi syndrome

## Abstract

Continual advances in our understanding of the human genome have led to exponential increases in known single nucleotide variants. The characterization of each of the variants lags behind. For researchers needing to study a single gene, or multiple genes in a pathway, there must be ways to narrow down pathogenic variants from those that are silent or pose less pathogenicity. In this study, we use the *NHLH2* gene which encodes the nescient helix-loop-helix 2 (Nhlh2) transcription factor in a systematic analysis of all missense mutations to date in the gene. The *NHLH2* gene was first described in 1992. Knockout mice created in 1997 indicated a role for this protein in body weight control, puberty, and fertility, as well as the motivation for sex and exercise. Only recently have human carriers of *NHLH2* missense variants been characterized. Over 300 missense variants for the *NHLH2* gene are listed in the NCBI single nucleotide polymorphism database (dbSNP). Using in silico tools, predicted pathogenicity of the variants narrowed the missense variants to 37 which were predicted to affect NHLH2 function. These 37 variants cluster around the basic-helix-loop-helix and DNA binding domains of the transcription factor, and further analysis using in silico tools provided 21 SNV resulting in 22 amino acid changes for future wet lab analysis. The tools used, findings, and predictions for the variants are discussed considering the known function of the NHLH2 transcription factor. Overall use of these in silico tools and analysis of these data contribute to our knowledge of a protein which is both involved in the human genetic syndrome, Prader–Willi syndrome, and in controlling genes involved in body weight control, fertility, puberty, and behavior in the general population, and may provide a systematic methodology for others to characterize variants for their gene of interest.

## 1. Introduction

Multiple in silico tools exist for the analysis of single nucleotide variants, whose numbers have exponentially increased from ~52 million in 2017, to over 715 million in the latest release of Ensembl 2022 [1]. However, even with these tools, there are few pipelines to systematically analyze variants in a gene of interest. Different protein types (i.e., transcription factors versus cellular signaling proteins versus extracellular matrix proteins) may need to be analyzed through different pipelines for in silico characterization of their individual functions, yet current pipelines of tools are not yet sophisticated enough for these endeavors. In this analysis, we use freely-available online in silico tools to analyze missense variants in the neuronal basic helix-loop-helix transcription factor, with the goal of providing a template for developing a pipeline for pathogenicity analysis of other transcription factor proteins. 

It has been 30 years since the original cloning and identification of the nescient helix-loop-helix 2 transcription factor, *NHLH2* [2]. Much has been learned about the role of the gene and protein, especially with regards to its role in maintaining fertility and normal body mass, using mouse models, humans, and phylogeny. In a review published nearly 10 years ago, the authors pose five outstanding questions on *NHLH2*, including “Do human SNPs in *NHLH2* contribute to any human obesity, physical activity, or fertility phenotypes?” [3]. We now have that answer for several *NHLH2* SNPs. For example, a nonsynonymous single nucleotide polymorphism/variant (SNV) in *NHLH2* changing an alanine to a proline at position 83 (A83P, NC_000001.10:g.115838126 C>G, CRCh38.p12, chr 1) structurally changes NHLH2 protein as predicted by in silico analyses, and as shown using Western blotting [4]. This SNV was originally discovered in two individuals with obesity who had no other detectable variants [5,6]. The SNV has never been added to the NCBI SNP database, likely due to low frequency but is within the same codon as rs1368574494, which results in the same alanine changed to a threonine. 

We have recently added four more clinical variants in *NHLH2* to the ClinVar database [7]. These variants are linked to hypogonadotropic hypogonadism in humans [8]. In particular, a R79C variant (Variation ID 1326288) found in a consanguineous family completely inactivates the ability of NHLH2 transcription factor to bind to the *MC4R* promoter (as shown previously [9]), and transactivates the *KISS1* promoter (also as shown previously [10]). The 19-year-old Turkish man who was homozygous for this variant displayed not only hypogonadotropic hypogonadism, but early adolescent obesity (OMIM 162361, [8]). His parents and sister, who were heterozygous for the variant, did not demonstrate these phenotypes; indicating recessive inheritance.

In this article, we use variants in the *NHLH2* gene to provide a pathway for discovery of variants that have a high pathogenicity prediction, and which should be further analyzed in wet labs. To carry this out, more than 300 missense variants currently listed in the NCBI database for *NHLH2* were characterized using several different SNP prediction programs. Those with the strongest predicted pathogenicity were further analyzed with respect to whether the individual variant might affect tertiary structure, post-translational modification, nuclear/nucleolar localization, and phylogenetic conservation. The use of these in silico techniques on missense variants allows for sorting based on the predicted ability to inactivate NHLH2 function in DNA binding, and regulation through post-translational modifications. The 21 identified missense variants can then be further analyzed using lab-based techniques, so that genotype–phenotype predictions can be made for individual carriers of these variants.

## 2. Results

### 2.1. Chromosomal Location, mRNA Transcripts, Protein Structure, and Identification of Variants

The human *NHLH2* gene is located on chromosome 1p13.1 with the longest transcript (X1 variant) at 9782 bp long (Figure 1A). The gene is located in the complement orientation, according to the latest assembly of the human genome (annotation release 110, 2022; Assembly GRCh38.p14). Transcript variant 1 has three exons, and a 2512 bp linear RNA, while transcript variant 2 has two exons and is 2492 bp in length. All three transcripts contain the protein coding region of 408 nucleotides, which is contained within one exon (exon 3 for transcript variant 1 and exon 2 for transcript variant 2) (Figure 1B). The NHLH2 protein is 135 amino acids long, and shares C-terminus homology with basic helix-loop-helix transcription factors. Multiple gene regulatory targets of mouse *Nhlh2* have been characterized, and many are key players in the body weight, and hypothalamic–pituitary–gonadal axis (for a review, see [3]). 

There are currently 318 missense SNVs listed for the NHLH2 protein coding region in the NCBI dbSNP database [11]. Variants that only affected NHLH2 transcript variant X1 were removed from the list, as this transcript makes both the recognized protein, and another transcript that may code for protein, but is currently without biological confirmation. The resulting 109 missense mutations are found only in the coding region that is consistent between the three splice variants shown in Figure 1B. These variants were analyzed using the PROVEAN pathogenic variant analysis prediction program prior to decommission of the online site [12]. Seventy-three variants had PROVEAN scores higher than −2.5 and were eliminated on that criterion for the purpose of this study. The PROVEAN scores for these 73 variants ranged from 0.0003 to −2.113 and were predicted to be either tolerated, or with lower potential for pathogenicity. The remaining 37 variants (one with a single number, but two associated changes) had PROVEAN scores ranging from −2.648 to −8.367 and were considered pathogenic by this criterion. As the online version of PROVEAN was recently retired, the 37 variants were further analyzed using three additional tools: Mutation Assessor, release 3 [13], SNAP [14], and CRAVAT [15]. Using Mutation Assessor, the Functional Impact (FI) scores show that only 9 out of the 37 predicted pathogenic variants were considered “high” impact, and 16 out of 27 had medium impact (Appendix A). Using the two other tools, eight of the nine variants identified by Mutation Assessor were again considered deleterious. In addition, five of the variants identified by Mutation Assessor as low–medium impact were identified as higher impact by both SNAP and CRAVAT. Two variants that were given low impact scores by Mutation Assessor were high for only SNAP, and not CRAVAT, while three additional variants had medium scores for Mutation Assessor, and high scores for SNAP only. None of the SNVs were predicted to be neutral by SNAP2 (Appendix A). According to one comparative analysis of prediction tools, Mutation Assessor had one of highest accuracies (81%), combined with high specificity (86%) [16]. Both SNAP and CRAVAT (CHASM) tools were analyzed in this article as well, but PROVEAN was not included in the comparison. SNAP had relatively lower accuracy (68%), but similar specificity at (81%), while CRAVAT was highest at 89% accuracy and 99% specificity. A combined approach as carried out here provides additional insight into variants with the most deleterious effects on NHLH2 protein function.

As shown, and consistent with many missense variants, the frequency for each of these variants is generally low, ranging from 0 identified individuals in a dataset, to a high of 3/238512 (0.001%). The new release of the All of Us Research Program data through their genome variants database represents sequences from ~98,500 whole genome sequencing results, and ~165,000 genotyping arrays in the aggregated data from 168,080 participants (Accessed on 12 December 2022) [17]. These are likely to increase as more genome data from the 831,000 individuals registered for the program become available. Of the 37 variants in Appendix A, only 5 were found in the All of Us genome variants database, and these are also listed in the frequency columns. Currently, one is not able to obtain additional information on the carriers of these alleles, but it is expected that genotype and clinical, lifestyle, and other data being collected by the program may eventually be linked and available for mining by researchers.

The location of the 37 variants on the NHLH2 protein is shown in Figure 2. For this analysis, none of the variants with PROVEAN scores below −2.5 were found in the N-terminal domain of the protein. This is in contrast with our previous work which identified human carriers of A9L and V31M variants, which were shown to be defective in gene transactivation in HEK293 cells [8]. Variants in the current analysis were clustered in the basic region (9 variants affecting 6 amino acids), helix 1 (8 variants affecting 8 amino acids), loop (7 variants affecting 6 amino acids), and helix 2 (12 variants affecting 9 amino acids). In comparison with the data from Appendix A, many of the variants with high predicted pathogenic scores are found within the loop and second helix of the protein. 

The NHLH2 protein shows strong homology within the basic helix loop helix domain, as assessed by phylogenetic animals of a distinct set of vertebrates (Figure 3A). Orthologs of the NHLH2 protein are found in both vertebrate and invertebrate species. To date, 270 orthologs have been sequenced from jawed vertebrates (Gnathostomata), with 179 of these from mammals, 55 from birds, and 58 from turtles, alligators, and lizards/snakes combined. These are 6 amphibian orthologs of NHLH2, and 1 each in the lungfishes (*P. annectens*), and cartilaginous fishes (*A. radiata*), as listed in the NCBI orthologs database. Boney fishes are not included in the NCBI orthologs page, but a search reveals that both the common carp (*C. carpio*) and zebrafish (*D. rerio*) sequences are available through the HomoloGene database on NCBI. The zebrafish sequence for Nhlh2 is smaller at 122 amino acids, sharing only 79.2% homology with humans, mainly in the basic helix-loop-helix domain (position 66–122 in the zebrafish sequence), with only one conserved amino acid change at position 82 in the zebrafish sequence which occurs within the first helix of the protein. The N-terminal end of the zebrafish Nhlh2 protein has deletions and alternative amino acids in 50 of 65 amino acids [18]. Several invertebrate versions of the *NHLH2* gene exist as well, including *D. melanogaster* (fruit fly), *L. salmonis* (salmon louse), and *R. varioornatus* (waterbear tardigrade), which are actually all more homologous to the paralogous gene *NHLH1*, and exist as a single gene, rather than paralogues in the organisms.

Nuclear and nucleolar localization sequences are present in all protein sequences analyzed for NHLH2, spanning from amino acid 66–80 (nuclear localization sequences) and 61–82 (nucleolar localization sequence) (Figure 3B,C). These data predict that the NHLH2 protein would normally be present in the nucleus or nucleolus. While nuclear localization is consistent with NHLH2’s function in transcriptional regulation, NHLH2 has not been localized in the nucleolus. While initially the nucleolus was thought to be only a site for ribosome biogenesis, recent data have demonstrated a role for the nucleolus in sequestering proteins during certain stress responses, including nutrient deprivation, and cold/warm stress conditions [19]. Previous lab-based studies from our laboratory have shown that hypothalamic NHLH2 mRNA levels are reduced with food deprivation and cold exposure and increased with food return or rewarming [20,21,22], and the identification of a nucleolar localization signal in the NHLH2 protein suggests that we should examine protein localization during these and other stress responses. 

### 2.2. Predicted Effects of Variants on Protein Post-Translational Modifications

The MuSiteDeep PTM prediction software predicts alterations in several different types of PTM including phosphorylation, glycosylation, ubiquitination, palmitoylation, addition of hydroxyproline or hydroxylysine, SUMOlaytion, and methylation [23,24,25]. There are predicted phosphorylation sites: each of the 37 variants, as well as the normal sequence for NHLH2 listed in Table 1 underwent analysis using MuSiteDeep. MuSiteDeep checks for glycosylation, ubiquitination, SUMOlaytion, acetylation, methylation, palmitoylation, pyrrolidone carboxylic acid, and hydroxylation in a FASTA entered sequence. NHLH2 reference protein analysis yielded only phosphorylation, ubiquitination, pyrrolidone carboxylic acid modification, and glycosylation. As shown in Figure 4A, the NHLH2 protein has 10 predicted serine and one threonine phosphorylation site within the N-terminus of the protein, and none exist past the threonine site at amino acid 75. Variants marked with yellow triangles result in loss of phosphorylation at position 75. Two variants, rs75107396 and ra1194455186, lead to additional phosphorylation, at positions 108 (threonine) and position 65 (serine), respectively, as indicated by the light-yellow triangles. Both variants change the amino acids at those positions into amino acids with the potential for phosphorylation. A putative glycosylation site is lost at position 106 with a proline to serine change at position 105. The remaining predicted changes due to missense mutations result in the addition of post-translational modifications with an addition of hydroxylated residues at amino acids 124 and 125, due to the tyrosine to cysteine change from rs1433737875. Deacetylation of NHLH2 protein by SIRT1 deacetylase at lysine 49 was previously described [26]. However, MuSiteDeep did not detect lysine acetylation of the WT protein. One variant, rs1650931348, led to a change from a glutamine to a lysine residue which MuSiteDeep predicted would be acetylated. In addition, we had previously used a different in silico tool, the GPS-PAILS program http://bdmpail.biocuckoo.org/, accessed on 15 July 2022) [27] which had predicted acetylation of seven lysine residues on the WT NHLH2 protein, including the position 49 acetylation. 

### 2.3. Predicted Effect of Variants on Protein Tertiary Structure and Function

The basic region along with the first part of helix 1 of the helix-loop-helix family of transcription factors is known to contribute to DNA binding to an E-box motif (for a review, see [28]). Each transcription factor dimerizes with a partner, and contacts half of the E-box motif, which in the case of NHLH2 is most commonly “CAG”, using the 5th, 6th, 8th, 9th, and 13th amino acids as primary contacts with the DNA [29]. However, the NCBI Conserved Domain Search [30] predicts amino acids shown in yellow circles (Figure 5A) as those for the putative DNA binding sites. The variants in NHLH2 for this region are shown in Figure 5A (red hexagons). The remainder of helix 1 along with helix 2 (Figure 5B,C) interact with the dimerization partner along the predicted dimer interface. NHLH2 has a very short non-helical region immediately following helix 2 with unknown function. Tertiary structural analysis using the IntFOLD server was used to study any gross deformations of structure called by each of the 37 variants analyzed, as well as to analyze if the DNA binding domain was intact in the variant sequences.

As shown in Figure 5D, four variants resulted in loss of predicted DNA binding domain in its entirety, although these variants were with amino acids outside of the direct DNA binding domain (Figure 5A). Previous published work in our lab showed that the R79C variant found in a human with hypogonadotropic hypogonadism led to an alteration in the DNA binding amino acids from R79, H82, R85 to Y78, H82, E86 [8]. While we did not carry out a DNA-amino acid binding analysis with these 37 variants, Appendix A shows that modeling the NHLH2 protein with DNA for an additional 11 variants results in what appears to be an altered DNA binding structure for the protein (Appendix A). For example, Y78C and Y78H variants both appear to alter the tertiary model of NHLH2 bound to DNA, although the protein is still clearly interacting with DNA. More analysis, including wet lab experiments would be needed to confirm these predictions.

### 2.4. List of Most Pathogenic Variants, Predicted by In Silico Analysis

Using SNV pathogenicity tools, followed by specific analysis of possible alterations in protein post-translational modifications and since NHLH2 is a transcription factor, analysis of any changes in predicted DNA binding, a list of 21 variants (with 22 amino acid changes) for further lab-based studies were generated (Table 1). The in silico experiments initially examined 318 missense/nonsynonymous variants in the NHLH2 protein sequence. The 21 remaining variants for further investigation represent a 93% enrichment (just 6.6% of the variants are deemed significantly pathogenic for further analysis). In addition, laboratory experiments can be tailored to the predicted pathogenic consequence, such as DNA binding or changes in secondary modifications. The position of the variants along the 3D protein structure are shown in Figure 6.

## 3. Discussion

Polymorphisms in NHLH2 have been implicated in human hypogonadotropic hypogonadism with the associated phenotypes of low exercise and increased body weight [8]. These variants were identified by deep sequencing, but 318 other missense variants in the NCBI SNP database have not been further characterized in humans or in vitro. PROVEAN analysis of variants was used to initially identify 37 missense variants that were predicted to have functional consequences. These 37 variants were further analyzed using in silico tools, yielding 4 variants that are predicted to result in no DNA binding activity, and 12 with changes in predicted post-translational modifications. Interestingly, none of the variants overlapped with respect to prediction of DNA binding or post-translational modification changes. In addition, the variant analysis programs often differed in severity predictions. Only two variants, Y125C (rs1433737875) and K115N (rs1354640857), were predicted to be deleterious by all four single nucleotide polymorphism analyses and have an additional deleterious prediction (Y125C for additional of hydroxylation post-translational modification; and K115N for loss of predicted DNA binding ability). 

A total of 16 SNVs were identified for further wet lab analysis. Most of these SNVs are very infrequent in the databases, with only zero to one individual identified. These SNVs are similar in frequency to the three SNVs that we added to dbSNP recently which had never previously been characterized [8]. Interestingly, K115N which is considered deleterious by multiple criteria has the highest frequency with the alternate allele at 0.04% globally and in the European subgroup. This could suggest that carriers could number in the millions worldwide. 

Use of these in silico tools can help researchers to focus their wet lab research on variants with the highest predicted consequences. In addition, these tools can help identify amino acids that are key to protein function, and to then design experiments to directly test these effects. Variants in NHLH2 have been previously shown to affect DNA binding (R79C) [8], and gene transactivation (A9L, V31M, R79C, A83P) [4,8] in wet lab experiments. These types of experiments would be the next to be carried out for variants such as K115N and Y125C-containing proteins created by in vitro mutagenesis experiments. 

This set of studies did not use in silico protein interaction prediction tools as no bHLH proteins have been shown to experimentally interact directly with NHLH2. We and others have shown SP1 [26] and STAT3 [31] to form protein:protein interactions with NHLH2 in cell line-based studies, but these interactions were not replicated by the in silico tool used (PEPPI [32]). As new tools become available, and predictions are tested with wet laboratory analysis, the tools will become better at predicting dimer partners that may be in macromolecular complexes. In addition, variants in non-coding regions of the gene that could result in alternative splicing, mRNA stability, or translation were not analyzed in this study, although two of our previous studies have included non-coding variants from humans [4,33].

In summary, in silico-based analyses are tools that can inform future wet experiments, and also aid genetic therapists in determining if new variants have the potential to be disease causing. NHLH2 variants in humans are rare, but this does not diminish their importance for the individual carrier, and for the researcher who is dissecting the biological mechanisms of the protein. Future predictions in non-coding regulatory regions and in the *NHLH2* promoter region are also necessary to fully characterize upstream and downstream consequences of SNVs in *NHLH2*.

## 4. Materials and Methods

All of the work in this manuscript was performed using online databases and in silico web-based prediction programs.

### 4.1. Identification of NHLH2 Missense SNVs for Further Study 

Missense variants in the *NHLH2* coding region were identified using the search function in the dbSNP database (National Center for Biotechnology Information, Bethedsa, MD, USA) [11], and the gene name, *NHLH2*, along with the search filter “missense”. Population frequency data for each SNV were recorded and used to narrow down the dataset to 109 missense mutations. Each variant was further analyzed using the pathogenic prediction program PROVEAN (J. Craig Venter Institute, La Jolla, CA, USA) [12], prior to its retirement this year. Variants with scores less than −2.5 were selected for further analysis, resulting in 37 missense mutations in this category. 

Because PROVEAN has become outdated, the potential impact of each SNV was reassessed using three different scoring techniques. SNVs were first run through the Mutation Assessor server (Memorial Sloan Kettering Cancer Center, New York, NY, USA), which yields an FI score for each mutation. FI scores are ranked neutral (<0.85, green), low (0.85–1.9, blue), medium (1.9–3.5, yellow), or high (>3.5, red). In total, 5 of the 37 original deleterious SNVs were marked as low impact (green highlight) by Mutation Assessor, and 7 more SNVs were considered neutral (blue highlight) by FI scoring. The remaining 25 deleterious SNVs were confirmed by Mutation Assessor to likely have significant physiological impact.

After FI scoring was completed, the SNVs were run though the SNAP2 variant prediction software (Technische Universitat, Munich, Germany). And the CRAVAT analysis tool (Johns Hopkins University, Baltimore, MD, USA) The SNAP2 tool runs all possible single point mutations along a desired amino acid sequence and provides an impact score and accuracy percentage for each possible mutation [14,34,35]. The thousands of possible results from SNAP2 were filtered and the results for the 37 SNVs of interest were recorded. SNAP2 scoring yielded 16 SNVs with medium impact (score between −50 and 50, yellow highlight) and 21 SNVs with high impact scores (>50, red highlight). 

CRAVAT scoring was also performed on each of the 37 SNVs of interest. The CRAVAT server produces a VEST score with an associated p-value that is used to predict the physiological impact of single point mutations [36,37]. SNVs that received a VEST score over 0.745 (on a 0 to 1 scale) were predicted to have possible significant pathogenic impact. Only 14 of the original 37 deleterious SNVs had high VEST scores when analyzed via CRAVAT and were considered to have significant impact. 

### 4.2. Illustrator for Biological Sequences

The online program “Illustrator for Biological Sequences” (IBS) was used to draw NHLH2 protein and annotate variants on the protein (Figure 2) as well as the DNA binding domain and variants (Figure 5A). It can be found at http://ibs.biocuckoo.org/ (Cuchoo Group, Wuhan, China) [38].

### 4.3. Clustal Omega Phylogenetic Alignment Analysis

Phylogenetic alignment using the Clustal Omega Multiple Sequence Alignment program (European Molecular Biology Laboratory, Cambridge, UK) [39,40], and inputting the normal and variant human *NHLH2* sequences, along with the *Nhlh2* protein sequences from *Pan troglodytes* (chimpanzee), *Macaca mulatta* (Rhesus monkey), *Mus musculus* (mouse), *Bos taurus* (cattle), *Gallus gallus* (chicken), and *Danio rerio* (zebrafish), with the output as ClustalW with character counts, and all other settings were defaults for the server. 

### 4.4. Nucleolar and Nuclear Localization Signal Prediction

NP_005590.1 (human) and NP_848892.1 (mouse) amino acid sequences were used as inputs for the online sequence prediction programs. NLStradamaus was used for the nuclear localization signal sequence (University or Toronto, Toronto, Canada) [41], and NoD for nucleolar localization (University of Dundee, Dundee, Australia) [42].

### 4.5. Post-Translational Modification Prediction

Potential effects on PTM pattern were analyzed using MuSiteDeep post-translational modification software (University of Missouri, Columbia, MO, USA) [24]. Normal and variant *NHLH2* sequences were compared in the output from the in silico analysis. Any predicted alteration in PTM pattern from the normal protein was recorded. 

### 4.6. Protein Structure (2D and 3D) and DNA Binding Prediction

The NCBI Conserved Domain Search tool (National Center for Biotecnology Information, Bethesda, MD, USA) [30] was used to identify the predicted residues that were needed for DNA binding and dimerization. The IntFOLD server (University of Reading, Reading, UK) [43] was used to generate 3D structural models of the NHLH2 WT and missense variant proteins. Within the IntFOLD6 server interface, the FunFold2 server was used to predict the DNA binding domains for both the WT and variant proteins [44], and models were visualized using JMol (developed at the Minnesota Supercomputer Center, University of Minnesota, Minneapolis, MN, USA) [45].

### 4.7. PyMOL 3D Visualization of WT Structure

A PDB file containing data to create 3D rendering of the wild type NHLH2 protein was obtained from AlphaFold Protein Structure Database (European Molecular Biology Laboratory, Cambridge, UK) [46,47]. The data contained in the PDB file was uploaded into pyMOL (Schrodinger, Inc, New Yori, NY, USA) [48], a molecular visualization platform, to view the location of the 22 SNVs of interest listed in Table 1. Amino acid positions highlighted in yellow denote positions of variants that allow for DNA interaction that appears altered by FunFold2. The amino acids highlighted in red are predicted to have no DNA binding activity by FunFold2. The remainder of the protein is colored cyan for helical regions and rose for loops/disordered regions.

## Figures and Tables

**Figure 1 ijms-24-03193-f001:**
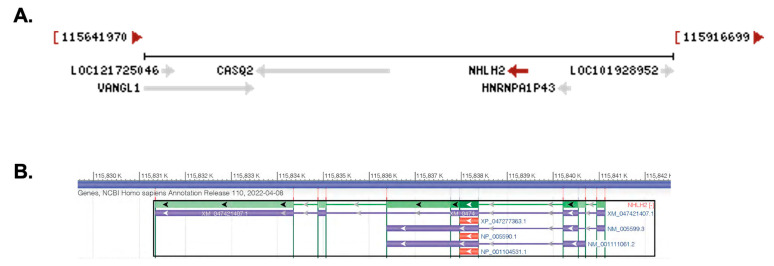
The *NHLH2* gene and transcripts. These pictures were captured using the NCBI gene database and genome view. (**A**) Chromosomal position and neighboring genes on chromosome 1p13.1. Arrows in grey or red (NHLH2 only) denote direction of gene transcription. Neighboring genes for *NHLH2* are shown using their abbreviated NCBI name. The base pair number range for this chromosome 1 segment is shown above the figure (**B**) *NHLH2* gene and related alternatively spliced transcripts. The protein coding region is shown in dark green in the main transcript, and as red for the cDNA sequence. All three transcripts appear to produce the same protein, with alternative 5’ and 3’ untranslated regions. Arrowheads indicate the direction of transcription.

**Figure 2 ijms-24-03193-f002:**
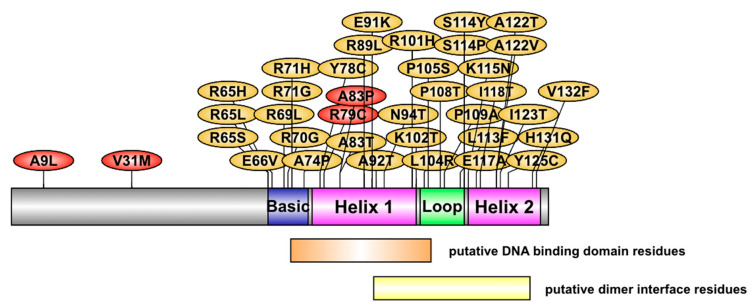
The NHLH2 protein and variant positions. A cartoon model of NHLH2 protein structure and the location of the variants studied in this article (yellow) and those previously reported (red). The position of the helix-loop-helix motif is based on previous studies. The position of the putative DNA binding domain residues and dimer interface residues is based on NCBI Conserved domain predictions. Single letter amino acids are used to indicate the amino acid in the reference protein, and the variant amino acid, separated by the position of the variant in the protein.

**Figure 3 ijms-24-03193-f003:**
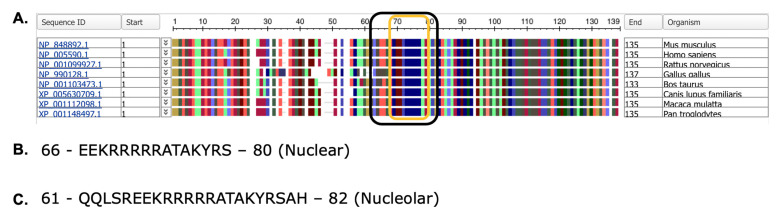
Phylogenetic analysis and nuclear/nucleolar prediction. (**A**) Phylogenetic analysis of NHLH2 protein sequences from *M. musculus* (mouse), *H. sapiens* (human), *R. novegicus* (rat), *G. gallus* (chicken), *B. taurus* (cattle), *C. lupus familiaris* (domestic dog), *M. mulatta* (rhesus macaque), and *P. troglodytes* (chimpanzee). The NCBI protein sequence ID numbers are given for each to the left. Amino acids are coded by color. (**B**) Nuclear localization signal as predicted by NLStradamus (**C**). Nucleolar localization signal as predicted by NoD. For B,C, single letter amino acid abbreviations are used.

**Figure 4 ijms-24-03193-f004:**
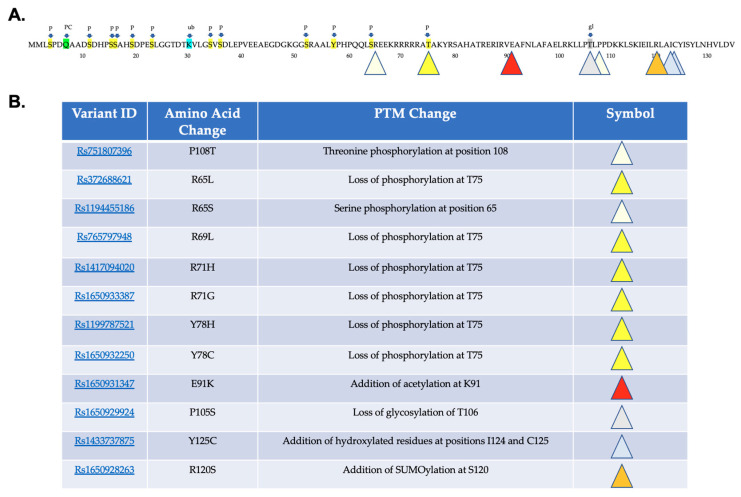
Post-translational Modification Analysis. MuSiteDeep was used to analyze the reference protein sequence for NHLH2. (**A**) Highlighted amino acids in sequence show post-translational modifications of P phosphorylation (yellow), PC (green) pyrrolidone carboxylic acid, ub (blue) ubiquitination, and gl (grey) glycosylation. Variant changes are indicated by the colored triangles that are categorized in (**B**).

**Figure 5 ijms-24-03193-f005:**
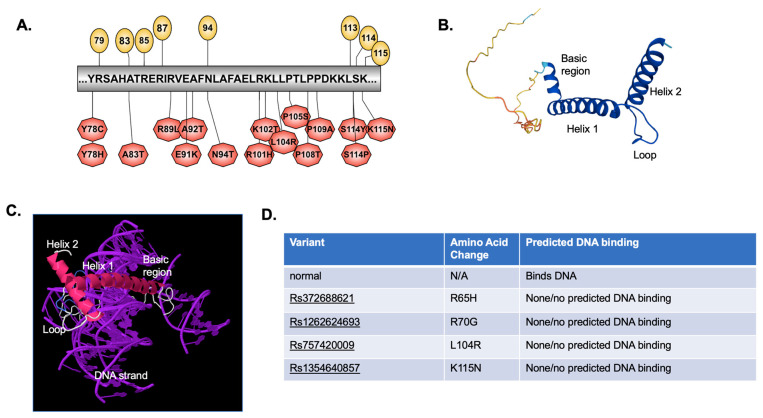
DNA binding motif and tertiary structure. (**A**) Illustrator for Biological Sequences was used to develop a cartoon of the predicted DNA binding region for NHLH2, and the location of variants within that region. (**B**) IntFOLD was used to analyze the tertiary structure of NHLH2 reference protein sequence alone and (**C**) bound to DNA (**D**) IntFOLD was also used to predict whether DNA binding occurred for each variant protein sequence modeled. The SNP variant ID number is provided, along with the position of the normal, and variant amino acid change for that variant. Additional tertiary structures of NHLH2 variants bound to DNA are found in Appendix A.

**Figure 6 ijms-24-03193-f006:**
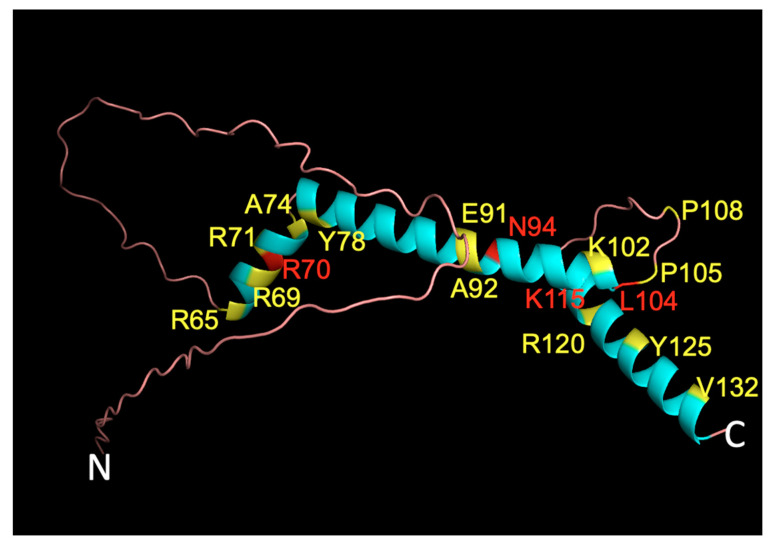
Three-dimensional NHLH2 structure overlayed with variant positions. The AlphaFold 3D model of human NHLH2 (HEN2, https://alphafold.ebi.ac.uk/entry/Q02577, accessed on 15 January 2022) was used as a backbone, and overlaid with colored amino acid positions representing to display residues where variants resulted in no DNA binding (red) or altered DNA binding (yellow). Seventeen positions, representing 21 variants (22 amino acid changes) are shown using the single letter code for the amino acid, the position of the amino acid in the protein, and the single letter code for the variant amino acid at that position. The N- and C- terminus of the protein is indicated.

**Table 1 ijms-24-03193-t001:** Predicted pathogenic variants for future study. Variant ID number from the SNP database is provided, along with the amino acid change using the single letter amino acid code and amino acid position number in the NHLH2 protein.

Variant ID	Amino Acid Change	Predicted Pathogenicity
Rs372688621	R65H R65L	Loss of DNA binding Loss of phosphorylation; Altered DNA binding
Rs1194455186	R65S	Additional phosphorylation; Altered DNA binding
Rs765797948	R69L	Loss of phosphorylation
Rs1262624693	R70G	Loss of DNA binding
Rs1417094020	R71H	Loss of phosphorylation
Rs1650933387	R71G	Loss of phosphorylation
RS772525034	A74P	Altered DNA binding
Rs1199787521	Y78H	Loss of phosphorylation; Altered DNA binding
Rs1650932250	Y78C	Loss of phosphorylation; Altered DNA binding
Rs1650931347	E91K	Additional acetylation; Altered DNA binding
Rs199738358	A92T	Altered DNA binding
Rs1352643678	N94T	Not predicted to binding DNA, but model could not be predicted
Rs781142041	K102T	Altered DNA binding
Rs757420009	L104R	Loss of DNA binding
Rs1650929924	P105S	Loss of glycosylation
Rs751807396	P108T	Additional phosphorylation
Rs1354640857	K115N	Loss of DNA binding
Rs1557829654	R120P	Altered DNA binding
Rs1650928263	R120S	Additional SUMOlaytion
Rs1433737875	Y125C	Additional hydroxylation
Rs1230535357	V132F	Altered DNA binding

## Data Availability

The data presented in this study are available in the Appendix A for this article.

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
