# Peer review of "In Silico Examination of Single Nucleotide Missense Mutations in NHLH2, a Gene Linked to Infertility and Obesity"

_ijms, 2023, doi:10.3390/ijms24043193_

Round 1

Reviewer 1 Report

In the present study, the authors have focused on the NHLH2 missense variants deposited in the dbSNP. They have extracted 318 missense SNVs and narrowed it down to 37 important SNVs by only considering SNV localization, in protein coding transcript, and functional effect by PROVEAN score >-2.5. They also checked homology of NHLH2 with basic helix loop helix domain with vertebrates. The impact of the selected SNVs were predicted on protein tertiary structure and function were also predicted using online servers. The results showed four variants no DNA binding activity, and 12 with changes in predicted posttranslational modifications. The authors proposed 16 important SNVs, which need to further analysis. The study only includes functional annotation of missense variants of NHLH2 using different basic bioinformatics tools.

1. The major limitation of the study is experimental validation. The author needs to validate at least one SNV experimentally.

2. The author may provide separate table for 16 significant SNVs.

Reviewer 2 Report

In this work, Madsen et al. described the analysis of missense variants which were predicted to affect NHLH2 function, and narrow down the results to 16 variants clustered around the basic-helix-loop-helix and DNA binding domains of the transcription factor for future downstream analysis. The design and analysis of the experimental results are well described, and I believe a few improvements in methodology and analysis workflow would make the conclusion more convincing and comprehensive.

1. For DNA binding predictions analysis, would it be possible to evaluate the impact of missense variants on known binding motifs for transcription factor NHLH2? For example, the JASPAR (PMID: 25045190) database has a large collection of known motifs from variant validation methods. I searched TF name NHLH2 in JASPAR and confirmed that known motifs for this TF are in the database. Also, the authors could consider using existing ChIP-Seq data that is publicly available to generate motifs for NHLH2 for such analysis.

2. I noticed the 3D structural models of the NHLH2 WT and missense variant proteins are generated from a web server which the latest server reference is in 2019. This is a little surprising to me since the AlphaFold, specifically AlphaFold2 which came out in 2021 has now become the dominant approach for structural prediction due to its significant improvements compared to any prior methods. The NHLH2 has already been added to the AF2 database: https://alphafold.ebi.ac.uk/search/text/NHLH2%20, while the structure prediction of variants may need to be carried out as additional work. While I understand AF2 may not work for the structural prediction of artificial proteins/variants. However, it is reported that AF2 works fine for cases in natural variants (PMID: 36418372). I believe the author could also consider model quality assessment methods and choose the final model from the best candidate from AF2 and existing predictions. 

Reviewer 3 Report

The paper suggests a computational analysis of 37 variations clustered around the transcription factor's basic-helix-loop-helix and DNA binding domains. This type of investigation could be useful in the future for directing a part of system experiments to evaluate the influence of variants on protein structures. The tools used provide significant evidence, even if the analysis might be improved by comparing more software. It is recommended that the Heatmap output from the SNAP2 software be included, as well as the revisions reported in the article. I also recommend that the table be better formatted.

Reviewer 4 Report

Dear authors,

The article "In silico examination of single nucleotide missense mutations in NHLH2, a gene linked to infertility and obesity” investigates the single nucleotide missense mutations in the NHLH2 gene using in silico tools. The article is helpful, with well-described methods and encouraging results.

However, I have some suggestions for the article that should be considered before publication:

Abstract:

The abstract is confusing, and I recommend that it be revised.

Introduction:

Please edit this part because it is difficult to read.

Results and Discussion

Table 1. Analysis of Pathogenic Missense Variants in NHLH2.

This table needs some adjustments:

In the first column (RS), please colour the name of the variants in black.

Instead of yellow, blue, green, and red highlights in the table, try underlining, bold, and italics.

I propose arranging the SNPs by Mutation Score, SNAP Score, or CRAVAT Score to make the table easier to read.

Overall, I appreciate the authors' efforts. The work is interesting, and the findings appear promising enough to serve as a starting point for further research.

Best regards,

Round 2

Reviewer 1 Report

The changes are made in the article. The experimental validation is a major limitation.